



# Measurement report: Long-term variations in carbon monoxide at a

# background station in China's Yangtze River Delta region

**Yijing Chen [1], Qianli Ma [2], Weili Lin [1,*], Xiaobin Xu[3], Jie Yao [2], Wei Gao [4]**
[1]College of Life and Environmental Sciences, Minzu University of China, Beijing 100081, China
[2]Lin'an Atmosphere Background National Observation and Research Station, Lin'an 311307,
Hangzhou, China
[3]Key Laboratory for Atmospheric Chemistry, Chinese academy of meteorological sciences, Beijing
100081, China
[4]Shanghai Key Laboratory of Meteorology and Health, Shanghai Meteorological Service, Shanghai
200030, China
*Corresponding author, email: linwl@muc.edu.cn




**Abstract**

This study analyzed the long-term variations in carbon monoxide (CO) mixing ratios from January
2006 to December 2017 at the Lin'an regional atmospheric background station (LAN; 30.3°N,
119.73°E, 138 m a.s.l.) in China's Yangtze River Delta (YRD) region. The CO mixing ratios were
at their highest ($0.69 \pm 0.08$ ppm) and lowest ($0.54 \pm 0.06$ ppm) in winter and summer, respectively.
The average daily variation of CO exhibited a double-peaked pattern, with peaks in the morning
and evening and a valley in the afternoon. A significant downward trend of $-11.3$ ppb/yr of CO was
observed from 2006 to 2017 at the LAN station, which was in accordance with the negative trends
of the average CO mixing ratios and total column retrieved from the satellite data (the Measurements
Of Pollution In The Troposphere, MOPITT) over the YRD region during the same period. The
average annual CO mixing ratio at the LAN station in 2017 was $0.51 \pm 0.04$ ppm, which was
significantly lower than that ($0.71 \pm 0.12$ ppm) in 2006. The decrease in CO levels was largest in
autumn (-15.7 ppb/yr), followed by summer (-11.1 ppb/yr), spring (-10.8 ppb/yr), and winter (-9.7
ppb/yr). Moreover, the CO levels under relatively polluted conditions (the annually 95th percentiles)
declined even more rapidly (-22.4 ppb/yr, $\alpha = 0.05$, r $= -0.68$) from 2006 (0.91 ppm) to 2017 (0.58
ppm) and the CO levels under clean conditions (the annually 5th percentiles) were relatively stable
throughout the years. The long-term decline and short-term variations in the CO mixing ratios at the
LAN station were mainly attributed to the implementation of the anthropogenic pollution control
measures in the YRD region and to the events like Shanghai Expo in 2010 and Hangzhou G20 in
2016. The decreased CO level may influence atmospheric chemistry over the region. The average
OH reactivity of CO at the LAN station is estimated to significantly drop from $4.1\pm0.7$ s$^{-1}$ in 2006
to $3.0\pm0.3$ s$^{-1}$ in 2017.

**Keywords:** CO, Long-term trend, Background level, the Yangtze River Delta region







## 1. Introduction


Carbon monoxide (CO) is a key intermediate in the atmospheric carbon cycle (Novelli et al.,
1992). In the troposphere, CO is one of the important air pollutants with high mixing ratios. The
volume mixing ratios of CO can reach an order of $10^{-6}$ (Khalil et al., 1999). CO is also a reactive
trace gas that considerably affects health, ecology, and climate, and hence recommended by the
Global Atmosphere Watch (GAW) of the World Meteorological Organization (WMO) for priority
observation. Fossil fuel combustion (mainly in the northern hemisphere), biomass combustion
(mostly in the southern hemisphere), and natural processes (the oxidation of organic compounds,
such as methane [$CH_4$] and isoprene) are the main sources of CO (Holloway et al., 2000; Thompson
et al., 1986; Novelli et al., 1998; Andreae and Merlet, 2001; Bakwin et al., 1994). The major sink
for CO is its reaction with OH radicals in the troposphere (Holloway et al., 2000; Thompson et al.,
1986; Novelli et al., 1998; WMO, 2003). The lifetime of CO in the atmosphere ranges from weeks
to months, which makes it an ideal tracer for atmospheric transport processes (Steinfeld and Jeffrey,
1998; Worden et al., 2013). Because $CH_4$ and CO can react with OH radicals (Thompson et al., 1992;
Daniel and Solomon, 1998), certain CO mixing ratios can indirectly cause a decrease in $CH_4$ and an
increase in $CO_2$. Therefore, CO is recognized as an important indirect greenhouse gas. Moreover,
CO is an important precursor for the photochemical generation of ozone in a polluted atmosphere
(Demerjian et al., 1972).
Continuous long-term observation is a method for studying large-scale CO sources, sinks, and
long-distance transport. This method allows the CO balance to be determined on a regional or global
scale (Fang et al., 2014). In the past decades, many studies have explored the long-term change in
CO levels through ground-, aircraft-, or satellite-based observations (Yurganov et al., 2010; Worden
et al., 2013; Ahmed et al., 2015; Cohen et al., 2018; Wang et al., 2018). Most of these studies have
revealed downward trends for CO concentration. For example, Yurganov et al. (2010) concluded
that the CO levels in the Northern Hemisphere decreased from July 2008 to January 2009, mainly
due to both the reduction in fossil fuel emissions and the presence of CO-poor air mass transported
from tropical regions. Worden et al. (2013) reported that the CO total column over China decreased





by 1.6% ± 0.5%/yr from 2002 to 2012. Ahmed et al. (2015) analyzed long-term CO observations at
two urban sites in Seoul and reported a downward trend of CO from 2004 to 2013. Wang et al. (2018)
found that from 1998 to 2014, the total column amount of CO over Beijing and Moscow decreased
at 1.14% ± 0.87%/yr and 3.73% ± 0.39%/yr, respectively. Cohen et al. (2018) analyzed the trends
of CO in the upper troposphere from 2001 to 2013. In their study, almost all observed trends were
negative, with the estimated slopes ranging from −1.37 to −0.59 ppb/yr. The CO data recorded in
the Arctic ice core indicated that the CO mixing ratios in this region decreased after the 1970s
(Petrenko et al., 2013).

Ground-based background measurements are crucial for verifying the accuracy of satellite

observation data, reflecting the impact of human activities on air quality and climate change, and
evaluating the effectiveness of pollution control measures. In China, many air pollutants have been
emitting in very large quantities. For example, the emission of CO was estimated to be about 171
Tg in 2010 (Li et al., 2017). To fight against the air pollution, the country has implemented a series
of emission control measures in the recent decade. The effectiveness of these measures needs to be
verified by observational data, in particular the data from background sites. Long-term background
observations over a decade are relatively scarce in China. Reports of long-term background
observations of CO are very limited in the literature (Meng et al., 2009; Liu et al., 2019; Zhou et al.,
2004; Zhang et al., 2011) and none of them present an analysis of CO variations over a decade. The
Yangtze River Delta (YRD) is one of the most developed regions in China. The long-term
observation of atmospheric background CO allows for a scientific understanding of the CO source
and sink cycle in this region. In this study, we present 12-year (from 2006 to 2017) ground-based
observations of CO at a background station in the YRD region. We analyze the long-term CO
variations and their determinants in the background areas of eastern China. The results of this study
function as scientific evidence for evaluating the effectiveness of pollution control policies and as a
reference for formulating practicable air pollution management and emission control measures.

**2. Monitoring sites and data collection**

The CO mixing ratios analyzed in this study were collected from January 2006 to December



2017 at Lin'an (LAN) station (30°25' N, 119°73' E, 138 m asl), a regional atmospheric background
monitoring site in China's Zhejiang province. The LAN station is one of the seven atmospheric
background stations operated by the China Meteorological Administration, and also a member
station of the World Meteorological Organization (WMO) Global Atmosphere Watch (GAW)
programme. The measurements at this station reflect the changes in the YRD region's atmospheric
background composition (Qi et al., 2012). The LAN station is located approximately 50 km west of
Hangzhou (the capital city of Zhejiang province) and 150 km southwest of Shanghai. It is influenced
by a typical subtropical monsoon climate. Fig. 1 displays the seasonal variations in air temperature
(T), wind speed (WS), pressure (P), and relative humidity (RH) as well as the wind direction (WD)
frequency at the LAN station from 2006 to 2017. These data were obtained from the regular
meteorological observations at the LAN station. As displayed in Fig. 1, the seasonal temperature
trend at the LAN station was of a convex shape. The highest and lowest temperatures occurred in
July (28.4°C) and January (4.1°C), respectively. In opposition to the seasonal change in temperature,
the seasonal change in atmospheric pressure at the LAN station showed a concave shape, with the
lowest and highest pressures occurring in July (989.51 hPa) and January (1010.81 hPa), respectively.
The seasonal patterns of the WS and RH at the LAN station were not as clear as those of air
temperature and pressure. The seasonal average WS was lowest in winter (1.9 m/s) and highest in
spring (2.1 m/s). The RH was highest in summer (77%) and lowest in spring (68%). The winds at
the LAN station mostly originated from the northeast and southwest, as shown in Fig. 1d. On
average, the northeast and southwest winds accounted for 29.2% and 22.6% of the winds,
respectively. The calm wind frequency was 4%.

A gas-filter correlation infrared absorption analyzer (48C trace level, Thermo Fisher, USA)

was used to measure the surface CO mixing ratios. The analyzer has a limit of detection of 0.04
ppm. Infrared radiation is chopped and passed through a rotating gas-filter lens, a one half of which
is filled with CO and the other with nitrogen. Thus, reference and measurement beams are produced
in alternation. The beams then pass through a narrow-band interference filter and sample cell.
Because the CO in the sample cell can only absorb the measurement beam, and the other gases can
absorb both beams, the measurement signal of CO could be obtained by comparing the attenuation
intensity between the reference and measurement beams.




The measurement signal from the CO analyzer was recorded every 5 min. Zero check and span
check were conducted every 6 and 24 hours, respectively. Multipoint (>5) calibration was performed
once a month using standard CO gas mixture (CO in nitrogen). Because the zero point of the
instrument drifted with time, we performed linear interpolation between two adjacent zero checks
to obtain the zero signals for given time point between the zero checks. These zero signals were
used in the corrections of the CO data. We performed response correction according to the results
of multipoint calibrations as well as the zero and span checks (Lin et al., 2009). Finally, we corrected
the data according to the quantity transfer and traceability results (Lin et al., 2011). Valid 5-minute
data were used to calculate the hourly mean mixing ratios. At least 10 data points were required for
any given hour to calculate that hour's mixing ratio. Missing data were caused by the malfunction
of the instrument from February 1 to 13, 2007, and from abnormal measurement fluctuations from
May 30 to July 17, 2009.

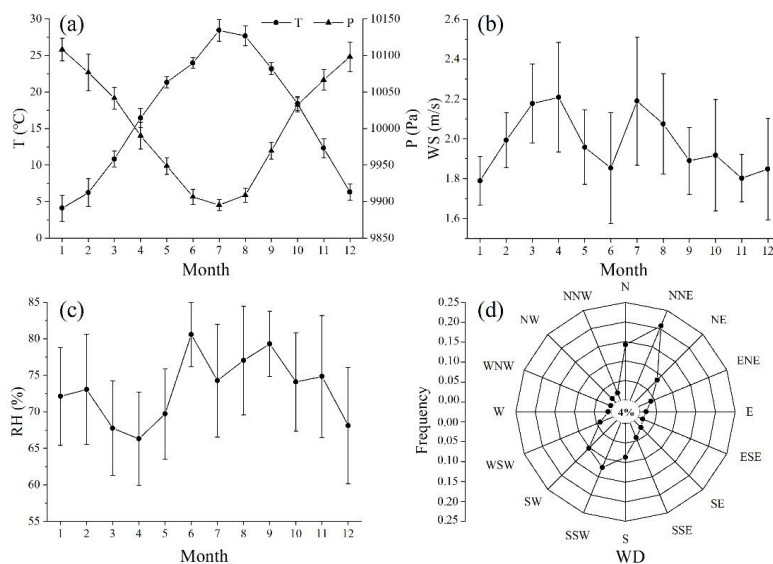

**Fig. 1.** Seasonal variations in (a) temperature, (b) WS, (c) air pressure, (d) RH, and (e) WD
frequency distribution (the static wind frequency was 4%) at the LAN station from 2006 to 2017
(an error bar represents one standard deviation)






## 3. Results and discussion

### 3.1 Observed levels and comparisons with other sites

Fig. 2 displays the time series of hourly mean CO levels at the LAN station from January 1, 2006, to December 31, 2017 and the linear fitting results of the hourly mean CO mixing ratios. The overall mean (±one standard deviation) and median values of the CO mixing ratios in the aforementioned 10 years were 0.62 (± 0.23) ppm and 0.57 ppm, respectively. The highest (2.98 ppm) and lowest (0.08 ppm) hourly mean mixing ratios occurred at 17:00 on January 10, 2008, and 18:00 on October 4, 2007, respectively. The highest hourly mean CO mixing ratio was considerably lower than the second-level hourly limit (approximately 8 ppm) of the ambient air quality standard in China (GB 3095-2012). The highest (2.38 ppm) and lowest (0.23 ppm) daily mean mixing ratios occurred on January 10, 2008, and August 31, 2011, respectively. The highest daily mean value was also below the daily limit for air quality standard (3.2 ppm). The lowest monthly average CO concentration was 0.39 ppm on August 2011, and the highest concentration was 1.00 ppm on January 2010. The median of daily mean CO levels from January 2006 to December 2017 was 0.58 ppm. The overall CO concentrations at the LAN were much higher than those observed at the Waliguan global baseline station from 2006-2017 and some regional background stations outside China (Table 1), indicating that East China has been one of the regions with high CO levels.

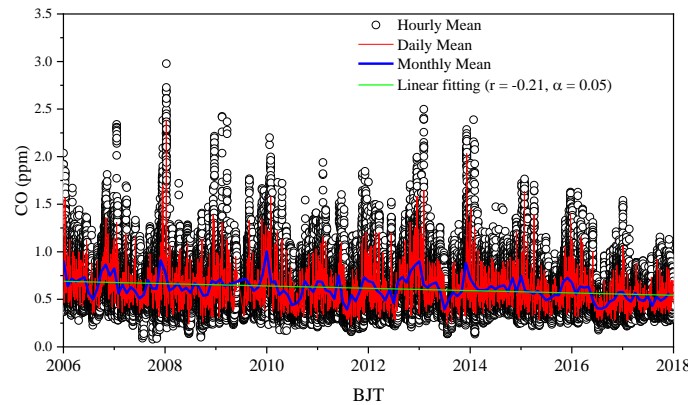

**Fig. 2.** Time series of the CO variations at the LAN station from 2006 to 2017



**Table 1.** Comparison of seasonal average CO variations at the LAN station and other similar background stations around the world

| Site | Location | Period | | | | | Trends (ppb/yr) | Ref. |
|---|---|---|---|---|---|---|---|---|
| | | | Spring (ppm) | Summer (ppm) | Autumn (ppm) | Winter (ppm) | | |
| Lin'an, China | 30°18'N, 119°44'E, 138 m | 2006~2009 | 0.65 ± 0.04 | 0.59 ± 0.04 | 0.65 ± 0.08 | 0.75 ± 0.05 | | This study |
| | | 2010~2015 | 0.59 ± 0.04 | 0.54 ± 0.06 | 0.62 ± 0.07 | 0.70 ± 0.07 | -11.3 | |
| | | 2016~2017 | 0.57 ± 0.08 | 0.46 ± 0.04 | 0.49 ± 0.03 | 0.56 ± 0.01 | | |
| Lin'an, China | 30°18'N, 119°44'E, 189 m | 2010/9~2012/2 | 0.47 ± 0.01 | 0.30 ± 0.01 | 0.41 ± 0.00 | 0.52 ± 0.01 | - | Fang et al., 2014 |
| Lin'an, China | 30°18'N, 119°44'E, 189 m | 2010/9~2017/5 | 0.38 ± 0.00 | 0.28 ± 0.00 | 0.37 ± 0.00 | 0.45 ± 0.00 | -16.3 | Liu et al., 2019 |
| Shangdianzi, China | 40°39'N, 117°07'E, 293 m | 2006/1~2006/12 | 0.75 ± 0.16 | 0.64 ± 0.14 | 0.80 ± 0.12 | 0.76 ± 0.13 | - | Meng et al., 2009 |
| Shangdianzi, China | 40°39'N, 117°07'E, 293 m | 2011/12~2017/5 | 0.16 ± 0.00 | 0.18 ± 0.00 | 0.14 ± 0.00 | 0.16 ± 0.00 | -1.3 | Liu et al., 2019 |
| Longfengshan, China | 44°44'N, 127°36'E, 311 m | 2006 | 0.21 | 0.20 | 0.27 | 0.38 | - | Wu et al., 2008 |
| Jinsha, China | 29°38'N, 114°12'E, 750 m | 2006/6~2007/7 | 0.44 | 0.39 | 0.66 | 0.60 | - | (Lin et al., 2011) |
| Waliguan, China | 36°28'N, 100°89'E, 3810 m | 2006/1~2017/12 | 0.13 ± 0.01 | 0.13 ± 0.01 | 0.12 ± 0.01 | 0.12 ± 0.01 | -0.67 | WDCGG |
| Tae-ahn Peninsula, Korea | 36.73'N, 126.13'E, 20 m | 2006/1-2017/12 | 0.27 ± 0.03 | 0.19 ± 0.04 | 0.21 ± 0.03 | 0.23 ± 0.02 | -0.43 | WDCGG |
| Yonagunijima, Japan | 24.47'N, 123.01'E, 30 m | 2006/1~2017/12 | 0.18 ± 0.03 | 0.09 ± 0.01 | 0.13 ± 0.02 | 0.19 ± 0.02 | -0.98 | WDCGG |
| Park Falls (WI), the U.S. | 45.93'N, 90.27'W, 868 m | 2006/1~2017/12 | 0.17 ± 0.02 | 0.16 ± 0.03 | 0.14 ± 0.02 | 0.16 ± 0.02 | -0.96 | WDCGG |
| Payerne, Switzerland | 46.81'N, 6.94'W, 490 m | 2006/1~2017/12 | 0.20 ± 0.04 | 0.14 ± 0.01 | 0.20 ± 0.04 | 0.28 ± 0.05 | -5.20 | WDCGG |

## 3.2 Seasonal variation


Fig. 3 shows the seasonal variations in CO mixing ratios at the LAN station and the number of
fire emissions (retrieved from the Global    Fire Emissions Database version 4 described in Werf
et al., 2017) in the YRD region (22°N~ 40°N, 112°E~123°E) from 2006 to 2017.

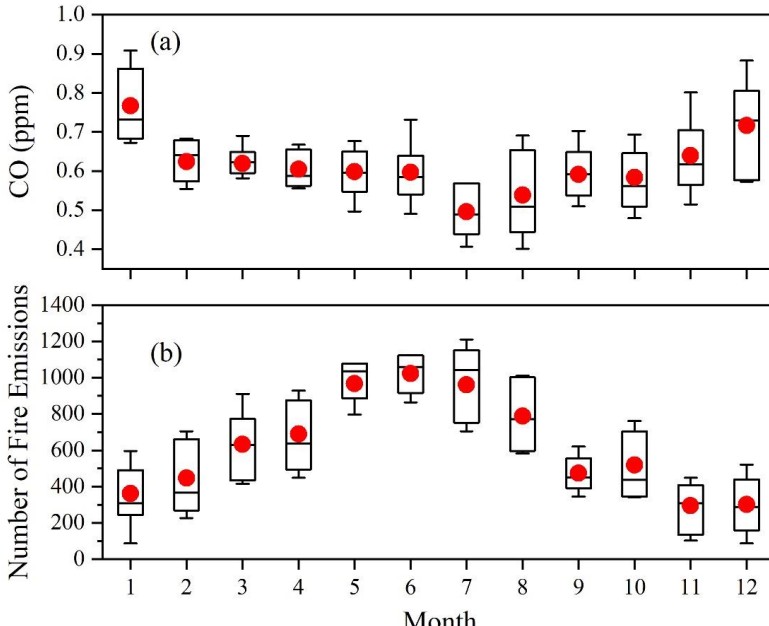


**Fig. 3.** Seasonal variations in CO mixing ratios at the LAN station and the number of fire emissions

in the YRD region from 2006 to 2017. The lines and dots in the box are the median and mean
concentrations, respectively, the box's lower and upper limits represent $25^{th}$ and $75^{th}$ percentiles
concentrations range, respectively, and the lower and upper whiskers correspond the $10^{th}$ and $90^{th}$
percentiles values.

As can be seen in Fig. 3(a), the average CO mixing ratios were the highest in the winter (0.69

$\pm$ 0.08 ppm), followed by the spring (0.61 $\pm$ 0.05 ppm), autumn (0.61 $\pm$ 0.09 ppm), and summer
(0.54 $\pm$ 0.06 ppm). In the winter, because of the weak radiation, the photochemical consumption of
CO in the atmosphere decreased. Also, the atmospheric stability was high and the diffusion
conditions were unfavorable. Therefore, atmospheric CO accumulated easily and reached its
maximum concentration in the winter. Nevertheless, the photochemical reaction was strong in the


summer, which resulted in an increase in the mixing ratios of OH radicals and the chemical
consumption of atmospheric CO. Moreover, the boundary layer height was relatively high in the
summertime, which promoted the vertical diffusion and dilution of CO in the atmosphere. Therefore,
the CO mixing ratios were the lowest in the summer. By contrast, the seasonal variations in the
number of fire emissions in the YRD region (Fig.3b) were opposite to the trend of the CO mixing
ratios in different months, which indicated that open fire burning was not a main factor affecting the
atmospheric CO concentrations at the LAN station from 2006 to 2017.
**3.3 Diurnal variation**

The daily variations in the CO mixing ratios were influenced by emission sources, atmospheric

transport (horizontal and vertical), and the evolution of the atmospheric boundary layer (Xue et al.,
2006). Fig. 4 displays the average daily variations in the CO mixing ratios at the LAN station, along
with those cities Shanghai, Nanjing and Hangzhou. As displayed in Fig. 4, the CO mixing ratios
exhibited double peaks, with higher CO levels in the morning and evening but lower CO levels in
the afternoon. The peak of the CO mixing ratios at the LAN station mostly occurred in the morning
(7:00–10:00) and at night (19:00–24:00). The lowest CO mixing ratios were observed between
12:00 and 16:00. The hourly CO mixing ratios usually reached their minimum value in the afternoon
due to the high atmospheric boundary layer, intense vertical diffusion mixing, and sufficient OH
radicals at that time. The peak CO mixing ratios at the LAN station occurred during the morning
and evening rush hours. This is consistent with those observed in the urban areas of Shanghai (Gao
et al., 2017), Nanjing (Huang et al., 2013a), and Hangzhou (Zhang et al., 2018) (Fig. 4). Thus, the
CO mixing ratios at the LAN station were affected by the pollutant emissions related to
transportation in the surroundings. However, the peak-valley difference of CO at LAN was much
smaller than those found in the cities, reflecting reduced impacts from direct emissions on this
background site.

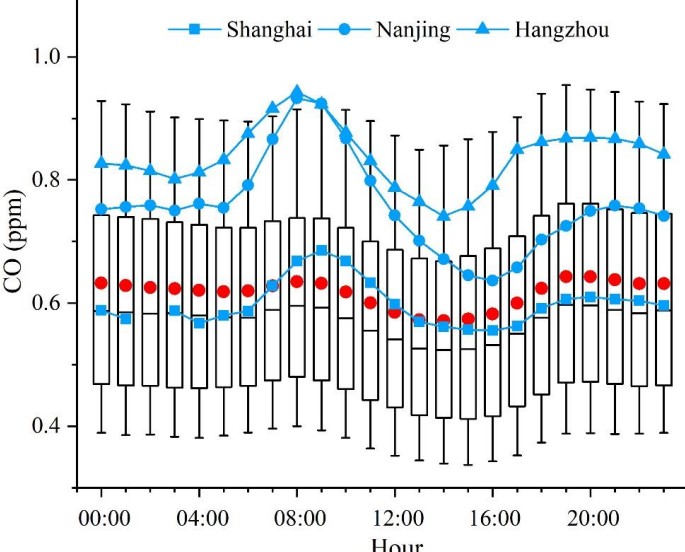

**Fig. 4.** Diurnal variations in CO mixing ratios at Shanghai, Nanjing, Hangzhou, and the LAN station
from 2006 to 2017. The lines and red dots in the box are the median and mean CO concentrations
at the LAN station, respectively, the box's lower and upper limits represent 25[th] and 75[th] percentiles
concentrations, respectively, and the lower and upper whiskers correspond the 10[th] and 90[th]
percentiles values.

**3.4 Long-term trends**
**3.4.1 Trends of annual means**
Fig. 5 shows the change in the annual mean CO mixing ratios at the LAN station from 2006 to
2017. The CO levels varied across the years. The World Expo was held in Shanghai from May to
October 2010, when air pollution prevention and control measures were strengthened in Shanghai
and its surrounding areas. Because of these strengthened measures, the number of days with good
air quality reached its highest value since 2001 (Huang et al., 2013b). Fig. 5 also indicates that the
average CO mixing ratio in 2010 was lower than those from 2006 to 2009 (1.5 months of data were
missing for the summer of 2009). The CO level continued to decline in 2011 but increased in 2012,
after which the CO level decreased steadily. China officially implemented the Action Plan for The
Prevention and Control of Air Pollution in 2013, which comprehensively intensified air pollution
control efforts and reduced multi-pollutant emissions. The plan called for 5-year efforts to improve





overall air quality and significantly reduce heavy pollution. As illustrated in Fig. 5, the effects of the
aforementioned action plan began to be observed in 2014, and the CO mixing ratios started to
decline significantly. Overall, the annual average of CO at LAN showed a decrease trend of 11.3
ppb/yr ($\alpha = 0.05$) during 2006-2017. For the period 2010-2017, we obtained a trend of -14 ppb/yr
($\alpha = 0.05$). This rate of decline in the CO mixing ratio was slightly lower than that ($-16.3$ ppb/yr)
reported by Liu et al. (2019) for the same station for 2010-2017. The measurements of Liu et al.
(2019) were performed using a cavity ring-down spectrometer, their air samples were drawn from
a tower (intake height: 50 m agl), and their trend was based on non-linear fitting on CO values after
removing those impacted by local events. The CO decreasing trend obtained in this study is smaller
than those reported by Ahmed et al. (2015) with values of $-20$ ppb/yr and $-13$ ppb/yr respectively
for two urban sites in South Korea during 2004–2013, larger than that reported by Liu et al. (2019)
with a value of $-1.3$ ppb/yr for a regional atmospheric background station in northern China during
2011–2017, and about a factor of 2-26 of those found in regional atmospheric background stations
in Korea, Janpan, and Switzerland (Table 1).

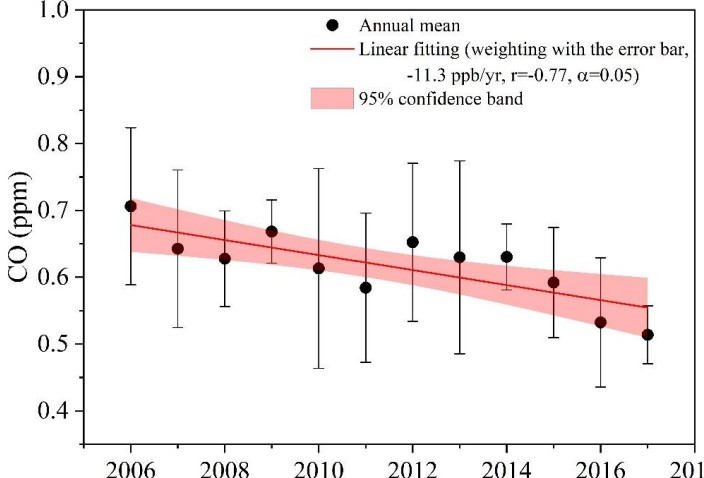


**Fig. 5.** Variation in the annual mean CO mixing ratios at the LAN station from 2006 to 2017 (the
error bars represent one standard deviation)
Considering the variation trend in Fig. 5 and the major air pollution control policies adopted
during the study period, we divided the study data into three subsets of data (collected during 2006–
2009, 2010–2015, and 2016–2017, respectively). The frequency distributions of average daily CO





mixing ratios in the three data subsets and the Lorentz curve fitting results are displayed in Fig. 6.
Approximately, a unimodal structure of CO frequency distribution was observed for all the datasets.
The peak values of the Lorentz curves can be used to characterize the background concentration
levels of atmospheric pollutants for a specific time and region (Lin et al., 2011). The peak of the CO
Lorentz curve shifted towards lower mixing ratios over time and the trailing phenomenon of the
fitting curve diminished gradually. The peak concentration of the fitting curve was $0.59 \pm 0.01$ ppm
from 2006 to 2009. During 2010–2015 and 2016–2017, the peak CO concentrations were $0.56 \pm$
$0.01$ and $0.49 \pm 0.01$ ppm, respectively. The peak frequency of the Lorentz curve was higher in
2016–2017 than in 2006–2015. Moreover, the peak width was significantly narrower in 2016–2017
than in 2006–2015. These are resulted from a decrease over time in the regional background mixing
ratios of CO.

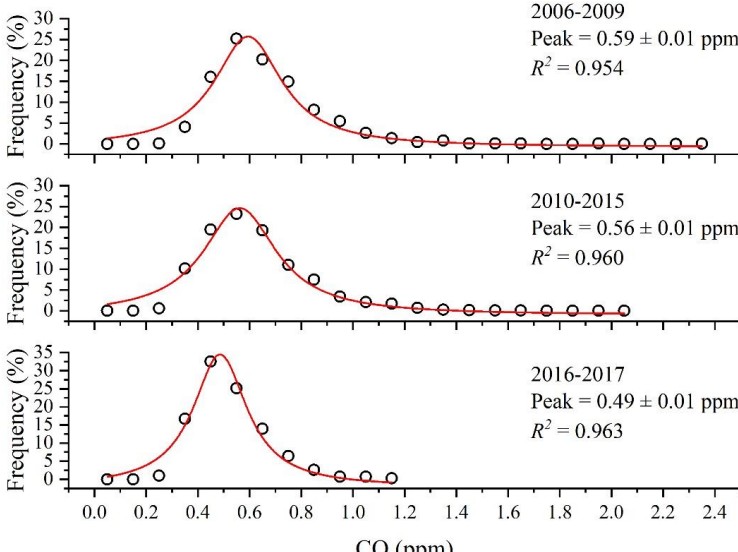


**Fig. 6.** Frequency distribution of the CO mixing ratios and Lorentz curve fitting results for
different time intervals
**3.4.2 Trends of seasonal means**

The time series of seasonal average levels of CO at the LAN station from 2006 to 2017 are

displayed in Fig. 7. Linear trends were calculated from the seasonal data, with standard errors being
used as weighting factors. From 2006 to 2017, the seasonal CO mixing ratios exhibited larger



fluctuations; nevertheless, an overall significant ($\alpha = 0.05$) decreasing trend was observed in each
season. The largest decrease (the slope of linear fitting) in the seasonal CO levels occurred in autumn
($-15.7$ ppb/yr), followed by summer ($-11.1$ ppb/yr), spring ($-10.8$ ppb/yr), and winter ($-9.7$ ppb/yr).

Table 1 presents a comparison of the seasonal average CO mixing ratios at the LAN station

and other background stations in the world from 2006 to 2017. As indicated in Table 1, the CO
mixing ratios at the LAN station in the four seasons between 2016 and 2017 were lower than those
between 2006 and 2015, with the largest decrease of 0.19 ppm occurring in winter. The seasonal
CO mixing ratios at the LAN station were marginally lower than those at the Shangdianzi station in
northern China (Meng et al., 2009), but were almost 3 times higher than those at many other regional
atmospheric background stations outside China, such as the Tae-ahn Peninsula station in Korea,
Yonagunijima station in Japan, Park Falls (WI) station in the U.S., and Payerne station in
Switzerland from 2006 to 2017 (Table 1). Moreover, the CO mixing values observed at the LAN
station were nearly 5 times higher than those observed at the Waliguan station, a global baseline
station in China. In conclusion, the CO levels at the LAN station were relatively high compared to
other regional atmospheric background stations outside China because of more intense
anthropogenic emissions in the YRD region.

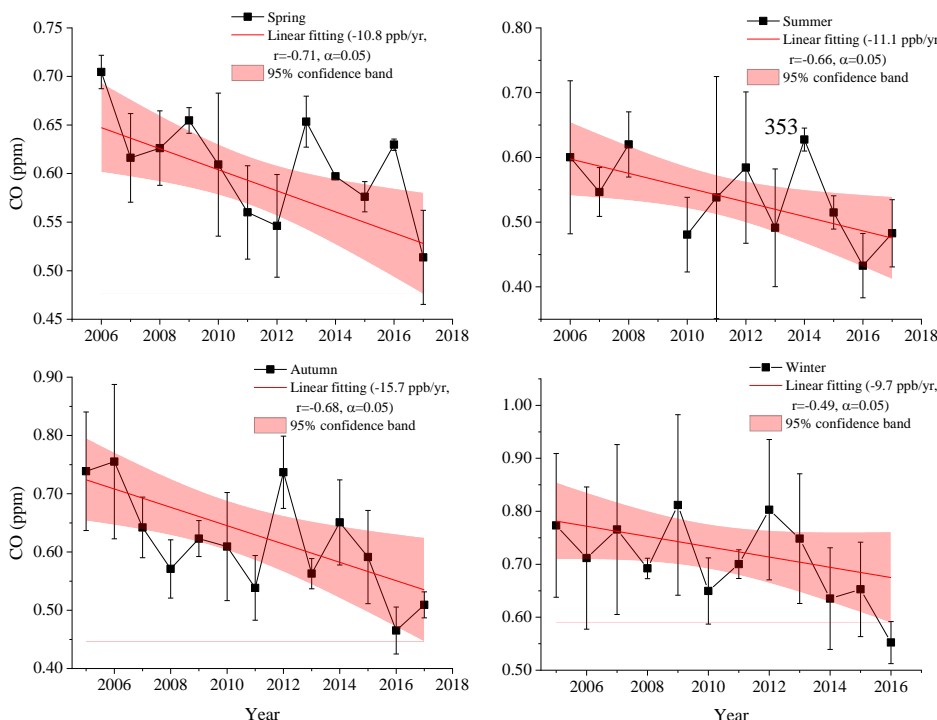


**Fig. 7.** Seasonal time series and linear fitting of CO mixing ratios at the LAN station

(Spring: March to May, Summer: June to August, Autumn: September to November, and Winter:

December to February)

### 3.4.3 Trends of CO levels under clean and polluted condition

In the annual statistics, the 95th and 5th percentiles of the CO mixing ratios can be viewed as

the CO levels in the most polluted and clean (background) airmasses, respectively. Here, we use

these two quantities to study CO trends under polluted and clean conditions, respectively, at the

LAN station. As illustrated in Fig. 8 (a), the CO concentration under the polluted condition

experienced an significant decreasing trend of -22.4 ppb/yr ($\alpha = 0.05$, r = −0.68) from 2006 (0.91

ppm) to 2017 (0.58 ppm) and that under the clean condition was relative stable (no significant trend)

throughout the years. This suggests that the CO levels in pollution plumes, which are highly

impacted by anthropogenic emissions in the YRD region, have been reduced greatly, while the

background levels of CO at the LAN station have not changed much. Fig. 8 (b) shows the average

CO concentrations from prevailing (N, NNE, NE, S, SSW and SW) and other wind directions. As



can be seen in Fig. 8 (b), the annual CO levels from different wind directions generally presented
similar patterns and all of them exhibited a significant ($\alpha = 0.05$) downward trend, suggesting that
the CO concentrations in the provinces and cities surrounding the LAN station have all decreased.

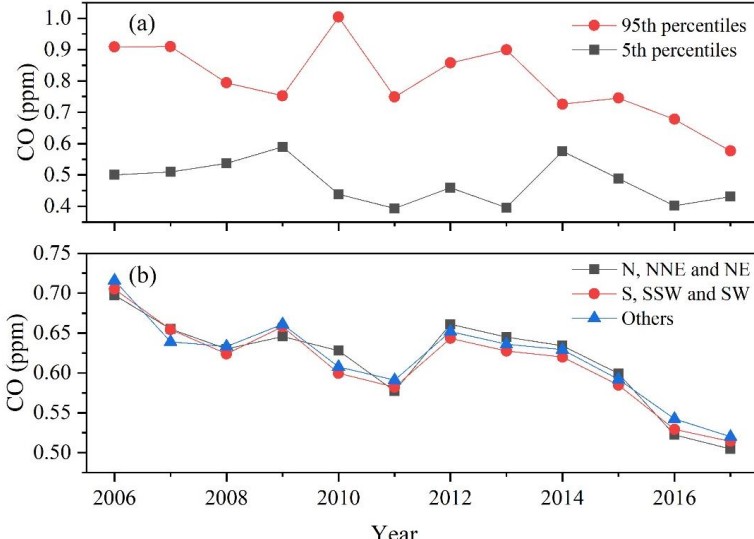


**Fig. 8.** Trends of CO mixing ratios at 95th and 5th percentiles and from different wind directions

**3.5 Causes and implications of the long-term variations**
**3.5.1 Impacts of Shanghai Expo and G20 in Hangzhou**

During the Shanghai Expo in 2010 (from 1 May to 31 October) and Hangzhou G20 in 2016

(from 24 July to 6 September), the Chinese government has implemented a series of joint pollution
control measures in the cities of the YRD region to ensure good air quality during these mega-events.
A satellite-based study (Hao et al., 2011) reported that a 12% reduction of CO concentration was
observed over Shanghai city during the Expo compared to the past three years. Zhang et al. (2017)
found that the ground CO levels in Hangzhou city decreased by 56% during G20 as opposed to
those in 2015. In order to further evaluate the effect of these control strategies, we compared the
annual trends of CO concentrations at the LAN station during the same period of Shanghai Expo
and Hangzhou G20, which are shown in Fig. 9 (a) and (b), respectively. The concentration of CO at
the LAN station was 0.54 ppm during the Expo and 0.41 ppm during the G20, and the values were
lower than those observed in Shanghai city (0.86 ppm) and Hangzhou city (0.53 ppm) in the same
period. Sharp decreases (reductions of 18 % during the Expo in 2010 and 35% during the G20 in
2016) of the CO mixing ratios were observed at the LAN station compared to those during the same




periods in the previous years, indicating that the pollution control measures worked well so as to
reduce atmospheric CO concentrations in the YRD region.

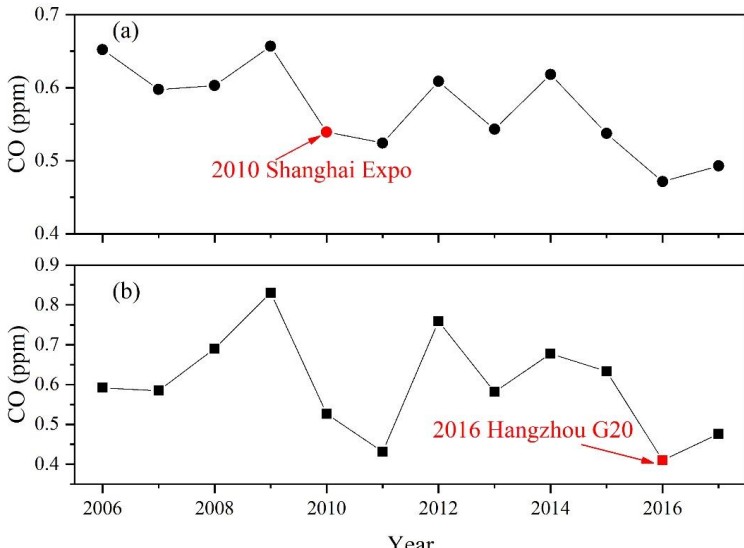


**Fig. 9.** Average CO levels for the periods corresponding to (a) 2010 Shanghai Expo (from 1 May
to 31 October) and (b) 2016 Hangzhou G20 (from 24 July to 6 September)


**3.5.2 Relationships with meteorological conditions**
Atmospheric CO mixing ratios are not only affected by local emission sources and the mixing
ratios of atmospheric OH radicals but also by meteorological conditions. Temperature, WS, WD,
and other meteorological conditions directly affect atmospheric stability and photochemical reaction
intensity, which influence the transmission capacity, generation, consumption rate, and lifetime of
atmospheric CO diffusion (Steinfeld and Jeffrey, 1998). Meteorological conditions varied across the
years of our study period. Such variations affected the comparison of the atmospheric CO mixing
ratios between different time intervals, especially when analyzing or evaluating the effectiveness of
pollution control policies. To minimize the effects of meteorological conditions on the analysis
results, we took temperature, WS, and WD as classification variables and analyzed the variation in
the CO mixing ratios under similar meteorological conditions during the three periods. The results
are displayed in Fig. 10.

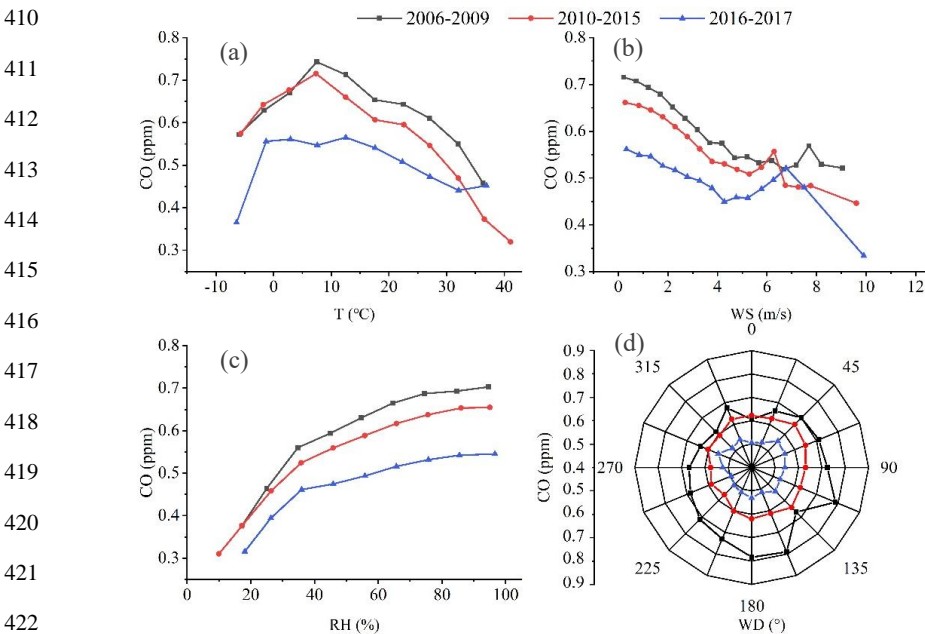


**Fig. 10.** Variations of CO mixing ratios in different periods with respect to temperature, WS, RH, and WD

As displayed in Fig. 10(a), the plot of the CO mixing ratios versus the temperature had a convex shape, with relatively low concentrations occurring at both high and low temperatures. Generally, because the photochemical reaction of CO intensifies at extremely high temperatures, and strong winds occur at extremely low temperatures, both high temperatures and strong winds can cause low CO mixing ratios. The decrease in the CO mixing ratios in a relatively high-temperature range during 2016—2017 was lower than the corresponding decreases in previous years. This result might be attributable to the summertime increase in energy consumption from the widespread use of air conditioners in China. Compared with 2006—2015, the stable area with high CO mixing ratios started to appear at lower temperatures during 2016—2017, which reflected the effectiveness of pollution control measures on the large emission sources. As displayed in Fig. 10(b), as the WS increased within a given range, the CO mixing ratios gradually decreased because of the strengthened diffusion and dilution of the atmosphere. When WS increased to a given level, where this level differed between the time intervals and continually decreased overtime, the CO mixing ratios increased with WS. This may be attributable to the pollution sources being increasingly close





to the LAN station because of increased urbanization over time. At a WS of 6–7 m/s, the CO mixing
ratios in the different time intervals tended to be consistent. As the WS continued to increase to
approximately 8 m/s, the atmospheric CO mixing ratios significantly decreased with the WS. As
displayed in Fig. 10(c), the CO mixing ratios correlated positively with RH, which is consistent with
the results reported by Turkoglu et al. (2004) and Ye et al. (2008). The main sink of CO is the
oxidation reaction with OH radicals (Steinfeld and Jeffrey, 1998). Because water vapor is a
precursor of clouds, at higher levels of RH, the atmosphere is more likely to be oversaturated with
water and form clouds, and, because clouds can reflect sunlight and reduce the ultraviolet radiation
reaching the ground, the photochemical reaction between CO and OH radicals is weakened (Ye., et
al., 2008). Fig. 10(d) displayed the change in CO mixing ratios with respect to WD. The figure
indicates that CO levels were the highest in the south sector of the LAN station.

Table 2 summarized the average percentage decrease in the CO mixing ratios during 2010–

2015 and 2016–2017 relative to CO mixing ratios in the previous time intervals under the same
meteorological conditions (temperature, WS, RH, and WD). As indicated in Fig. 10 and Table 2, the
CO mixing ratios during 2016–2017 were generally lower than those during 2006–2009 and 2010–

2015.


**Table 2.** Comparison of the average percentage decline in CO mixing ratios during 2010–2015
and 2016–2017 relative to CO mixing ratios in previous time intervals under the same
meteorological factors

|  | Decreased Percentage (%) | | | |
| --- | --- | --- | --- | --- |
|  | T | WS | RH | WD |
| 2010-2015[*] | -6.2 | -13.6 | -9.6 | -11.9 |
| 2016-2017[**] | -14.5 | -10.7 | -11.7 | -14.2 |
| 2016-2017[*] | -19.8 | -16.5 | -20.4 | -24.4 |

*: compared with 2006–2009, **: compared with 2010–2015.

**3.5.3 Changes in emissions in neighboring provinces**

China has implemented a comprehensive energy conservation and emission reduction policy

since 2006 (Zhao et al., 2008; Lei et al., 2011). Small and old factories and boilers have been



gradually replaced by larger and more energy-efficient alternatives. Although the focus of these
measures was to control sulfur dioxide emissions, these measures also greatly improved combustion
efficiency and thus decreased CO emissions (Zhao et al., 2012). Fig. 11 displayed the change in the
CO emissions in six provinces and cities around the LAN station from 2006 to 2017. The emission
data were obtained from the Multiresolution Emission Inventory for China (Li et al., 2017). As
indicated in Fig. 11, the average annual CO emissions of the provinces and cities surrounding the
LAN station declined significantly ($\alpha = 0.05$, r = −0.95), With an average decline rate of 170,000
tons/yr. The percentages of CO emission decreased during 2016–2017 in Shanghai city as well as
Jiangsu, Zhejiang, Anhui, Fujian, and Jiangxi provinces were −59.3%, −25.5%, −18.6%, −27.2%,
−40.1%, and −19.3%, respectively, relative to CO emission values during 2006–2009. In summary,
the decline in the CO mixing ratios at LAN station was mainly caused by the reduction of
anthropogenic CO emissions in its surrounding areas due to the effectiveness of the adopted
pollution control measures.

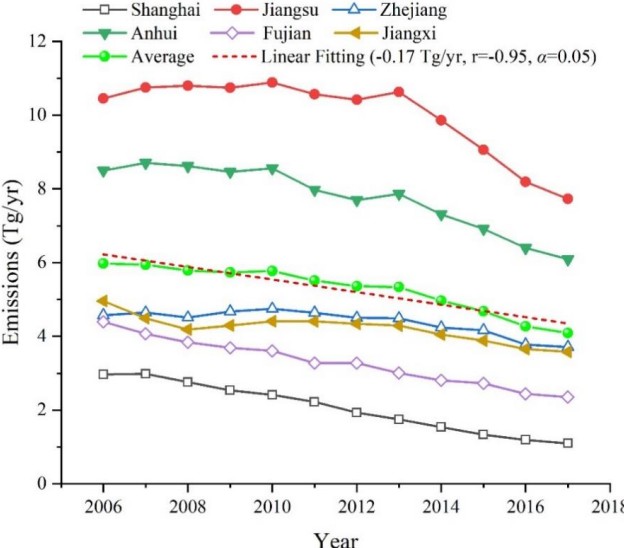


**Fig. 11.** CO emissions from 2006 to 2017 in the provinces and cities surrounding LAN station and
linear fitting of the average annual CO emissions of the six provinces and cities
Data source: http://meicmodel.org/dataset-mix.html
**3.5.4 Implications on regional atmospheric chemistry**





The tropospheric CO has been measured on a global scale from the Measurements Of Pollution

In The Troposphere (MOPITT) instrument on the spacecraft since 2000 (Deeter et al., 2017).

Monthly CO mixing ratios at the surface layer and the CO total column concentrations over the

YRD region from 2006 to 2017 were retrieved from MOPITT (MOP02J Version 8, 2018;

http://www.satdatafresh.com/CO_MOPITT.html). From 2006 to 2017, the average CO mixing ratio

from MOPITT over the YRD region (22.5°N~ 39.5°N, 112.5°E~123.5°E) in 2006 (0.11±0.02 ppm)

was higher than those in 2017 (0.10±0.02 ppm) , with a significant declining trend of -0.5 ppb/yr

($\alpha$ = 0.05, r = −0.82). As for the average CO total column, the value in 2006 ($1.91 \times 10^{18}$ ± 0.23

$\times 10^{18}$ molecules/cm$^2$) was also higher than those in 2017 ($1.76 \times 10^{18}$ ± 0.21 × 10$^{18}$

molecules/cm$^2$), with a significant declining trend of $-1.07 \times 10^{16}$ molecules/(cm$^2$·yr) ($\alpha$ = 0.05, r =

492       −0.70) from 2006 to 2017. They are in consistent with the negative trends of the aforementioned

ground CO levels measured in the sites of the WDCGG network (Table 1) and at the LAN station.

The major sink for CO is reaction with OH radical (Steinfeld and Pandis, 2006), so a decrease

in the CO concentrations may lead to an increase in the lifetime of OH radical and thus affect the

atmospheric OH photochemistry (i.e., ozone production). The lifetime of OH is defined as the

inverse of the OH reactivity (i.e., OH loss rates), and the total OH reactivity is calculated by summing

over all the products of the OH reactants (CO, volatile organic compounds, nitrogen oxides, etc.)

concentrations times their respective rate coefficients with OH ($k_{OH}$) (Kovacs and Brune, 2001; Di

Carlo et al., 2004). The lowest average total OH reactivity (5 s$^{-1}$~6 s$^{-1}$) observed in the rural areas

around the world (Ren et al., 2005; Ingham et al., 2009). The $k_{OH}$ of CO is 350 /(ppm·min) at the

standard temperature of 298K (Vukovich, 2000) and CO generally contributed 10%~20% to the

total OH reactivity at the rural sites of China (Lou et al., 2009). From 2006 to 2017, the average OH

reactivity of CO at the LAN station exhibited a significant downward trend of -0.07 s$^{-1}$/yr ($\alpha$ = 0.05,

r = −0.80) and the average monthly OH reactivity of CO dropped from 4.1±0.7 s$^{-1}$ in 2006 to 3.0

±0.3 s$^{-1}$ in 2017.

**4. Conclusion**

The average annual levels of CO at the LAN station during 2006–2009, 2010–2015, and 2016–

2017 were 0.66 ± 0.03 ppm, 0.62 ± 0.03 ppm, and 0.52 ± 0.01 ppm, respectively. From a seasonal



perspective, the highest seasonal average CO mixing ratio occurred in winter (0.69 ± 0.08 ppm),

followed by spring (0.61 ± 0.05 ppm), autumn (0.61 ± 0.09 ppm), and summer (0.54 ± 0.06 ppm).

The average daily variations in the CO concentration exhibited a double-peaked pattern, with high

CO concentrations in the morning and evening and low CO concentrations in the afternoon. Such

diurnal variations suggest that the CO mixing ratios at the LAN station were affected by traffic

pollutant emissions in its surrounding area.

The average annual atmospheric CO mixing ratios at the LAN station exhibited a significant

decreasing trend (−11.3 ppb/yr, α = 0.05) from 2006 to 2017, which was consistent with the negative

trends of the average CO mixing ratios and total column retrieved from MOPITT over the YRD

region. The measurements at the LAN station well reflected regional changes in atmospheric

background CO mixing ratios in the YRD region. The largest decrease in the CO level was observed

in autumn (-15.7 ppb/yr), followed by summer (-11.1 ppb/yr), spring (-10.8 ppb/yr), and winter (-

9.7 ppb/yr). The significant downward trend of the CO mixing ratios at the LAN station was not

caused by meteorological conditions but by strengthened pollution control measures, which

indicated that the adopted measures were effective. In spite of the nearly a quarter of reduction

during 2006-2017, the CO levels at the LAN station were still much higher those at other regional

atmospheric background stations around the world so that further reductions in CO emissions in the

YRD region are needed. The significant decrease of regional CO level has an implication for

atmospheric chemistry, considering the role of CO in OH reactivity. From 2006 to 2017, the average

OH reactivity of CO at the LAN station exhibited a significant downward trend of -0.07 s$^{-1}$/yr (α =

0.05, r = −0.80) and dropped from 4.1±0.7 s$^{-1}$ in 2006 to 3.0±0.3 s$^{-1}$ in 2017.

**Data availability.** The data sources of number of fire emissions, the annual CO emissions and the

CO concentrations retrieved from MOPITT over the YRD region are all listed in the reference, and

the CO concentrations and the meteorological data at the LAN station can be inquired about by

contacting the corresponding author.

**Author contributions.** YJC, WLL, and BXX developed the idea for this paper and formulated the

research goals. QLM and JY carried out the CO field observations at the LAN station. WG provided

the CO data in Shanghai. YJC and WLL wrote and revised the manuscript with contributions from

all co-authors.





**Competing interests.** The authors declare that they have no conflict of interest.
**Acknowledgments.** This study was funded by the National Key R&D Program of China
(2016YFC0201900), National Natural Science Foundation of China (91744206), and Beijing
Science and Technology program (Z181100005418016). We thank the personnel on duty at the LAN
station for their assistance. This manuscript was edited by Wallace Academic Editing.

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
