# Peer review of "Measurement report: Long-term variations in carbon monoxide at a"

_Atmospheric Chemistry and Physics, 2020_

## Referee Comment (RC1) · Anonymous Referee #1 · 23 Aug 2020

This manuscript reports a 12-years continuous measurement of CO at the GAW's regional atmospheric background station in eastern China (Lin'an). Temporal variations, especially the long-term trend, as well as the causes and implications of the CO decline were analyzed. Long-term observations of atmospheric compositions are crucial for understanding the variation trends of atmospheric chemical processes, but are relatively limited in China. The data presented in the present study are thus much valuable, and the data analysis and interpretation of results are fairly well. The organization and writing of the manuscript are also good. Hence, I would like to suggest that this manuscript can be accepted for publication after the following minor comments being properly addressed.

[Figure]

Specific Comments:

L62: from the perspective of atmospheric chemistry, intermediate is usually used to denote the 'product' of chemical reactions what may undergo further chemical reactions. So, I suggest the authors to change to another word, such as player?

L76-78: in general, the contribution of CO to ozone formation is not quite important in polluted atmospheres where the abundances of VOCs are high, but it may become important in the rural areas.

L84-85: the time period from July 2008 to January 2009 is too short to derive a "trend".

L99: have been emitted...

L114: Monitoring site...as you only have one site in this study.

L127-134: it would be better if the authors could provide the standard deviations for the average values (if any).

L140-141: rephrase "a one half of which is filed with CO and the other with nitrogen"

Fig. 1: the content in the figure is not consistent with that described in the figure caption. Please check and revise.

L240: delete the extra space between "Global" and "Fire"

L253: change "Nevertheless" to "In comparison" or "In contrast".

Fig. 4: please indicate the source of measurement data from Shanghai, Nanjing and Hangzhou. Are they also long-term data from 2006 to 2017?

L343-352: I suggest the authors to move this discussion (comparison with other sites) to Section 3.1.

Fig. 7: it is unclear why you show a number of "353" in the summer panel?

L389-392: a direct comparison in the average CO concentrations is quite rough. What

are the impacts from meteorological conditions on these differences?

L402: "transmission capacity" is hard to follow. What do you mean by this?

L425: showed a convex shape...

L466: Fig. 11 displays the change...

L470: with an average decline...

L500-501: rephrase this sentence.

Data availability: the measurement data must be made available for the research community. The authors are encouraged to deposit their data to an accessible repository.
* * *

---

## Referee Comment (RC2) · Anonymous Referee #2 · 4 Sep 2020

The manuscript presents an extensive overview of the long-term CO measurements at Lin'an in the Yangtze River Delta of China. The data cover 12 years and provide important information for the community to understand the changes and mechanisms in Chinese air pollution and its relationship with emissions, chemistry and meteorology. The data are valuable, and the analysis and writing is overall well done for a "Measurement Report". I recommend publication after the following issues are addressed.

The manuscript shows the variabilities/trends of CO at a variety of time scales of interest to scientists and policymakers. It would be very interesting and useful to compare the observed variabilities to emissions (both anthropogenic and natural) and to MO-

PITT data. Comparisons with emissions would reveal consistency and inconsistence between emissions and measured concentrations, and would offer insight into current limitations and strengths in emission data. At the present form, fire emissions are used to investigate the seasonality, and anthropogenic emissions are used to discuss the trends. These are very interesting, but examining other time scales would be insightful as well.

Also, comparison with MOPITT data at all time scales (in addition to trends) would provide further insight to the characteristics of CO over China and the representativeness of Lin'an measurements. For example, the trends in MOPITT agree in sign with that of Lin'an data, but to what extent is the two trends consistent quantitatively, and what are the implications of this consistency/inconsistency?

Diurnal cycle: the PBL mixing would be an important factor affecting the diurnal cycle. Could you show PBLH or other indicator of mixing (e.g., from assimilation data)? Also, there appears to be some phase shift in the peak mixing ratio between different sites. Could you comment on this?

Interannual variability: The variability is very large for CO mixing ratios, which may be related to meteorology and/or chemistry. Could you comment on this? What does this variability mean when relating CO concentrations to emissions? To what extent is the trend of CO consistent with those of emissions, guantitatively? And why?

Please define alpha for the first time (I assume it is the P-value). Also, why is alpha always equal to 0.05 throughout the text? Do you mean alpha < 0.05? Specifying the actual value of alpha would be better, given that the often mis-interpretation of the P-value, as discussed extensively in recent years.

Fig, 6. The choice of the three periods seems to be arbitrary – it is not obvious why the years can be grouped into these three periods. Could you show Lorenz curves year by year?

Fig. 8. There appears to be some non-zero trend in the 5% percentile data. Could you show the trend and discuss this in the context of the large-scale CO trend (e.g., shown in the literature)?

Fig. 9. The value in 2011 is also very low. Could you explain this?

Fig. 10. The analysis for Fig. 10 appears speculative. The apparent relationships are a result of processes at a variety of time scales, which render determinative explanations difficult. Also, the met fields used are correlated to each other, affecting the explanations specific to each met field.

Specific comments: L169, should be 12 years L526. "those" should read "than those" Table 1. The mixing ratio at Shangdianzi appears very low in 2011-2017. Could you check this? Fig. 3b. Do you mean the number of fire spots here? Fig. 5. Is the standard deviation calculated from monthly means? Fig. 7. Is weighting factors based on standard deviation of monthly mean values? Please clarify. Fig. 10 and Table 2. Please specify the intervals of each met field.

---

## Author Comment (AC1) · 20 Oct 2020

The authors' response and the measurement data (a excel file) was uploaded as a zip-file in the form of a supplement.

Please also note the supplement to this comment:
https://acp.copernicus.org/preprints/acp-2020-610/acp-2020-610-AC1-supplement.zip

---

## Author Comment (AC2) · 20 Oct 2020

We thank both referees for their very constructive comments and suggestions. We revised our manuscript according to their comments and suggestions and changes are highlighted in the revised version.

**Response to comments by referee 2**
**Anonymous Referee #2**
The manuscript presents an extensive overview of the long-term CO measurements at Lin'an in the Yangtze River Delta of China. The data cover 12 years and provide important information for the community to understand the changes and mechanisms in Chinese air pollution and its relationship with emissions, chemistry and meteorology. The data are valuable, and the analysis and writing is overall well done for a "Measurement Report". I recommend publication after the following issues are addressed.

The manuscript shows the variabilities/trends of CO at a variety of time scales of interest to scientists and policymakers. It would be very interesting and useful to compare the observed variabilities to emissions (both anthropogenic and natural) and to MOPITT data. Comparisons with emissions would reveal consistency and inconsistence between emissions and measured concentrations, and would offer insight into current limitations and strengths in emission data. At the present form, fire emissions are used to investigate the seasonality, and anthropogenic emissions are used to discuss the trends. These are very interesting, but examining other time scales would be insightful as well. Also, comparison with MOPITT data at all time scales (in addition to trends) would provide further insight to the characteristics of CO over China and the representativeness of Lin'an measurements. For example, the trends in MOPITT agree in sign with that of Lin'an data, but to what extent is the two trends consistent quantitatively, and what are the implications of this consistency/inconsistency?

Response: Thanks for your advice. In fact, we have tried to find the correlation between fire data and surface CO in biomass burning seasons, but no significant implications founded according to our current knowledge. We might find another way to deal with it and report it later. We added the correlation of annual surface CO with MOPITT CO data (see Fig. S1.). We found significant correlations ($p < 0.05$) between surface CO and MOPITT CO ($r = 0.75$ and $0.61$ for the MOPITT CO mixing ratio and total column, respectively) data, which indicate the good regional representativeness of Lin'an measurements. Although the negative trends were found both in surface and MOPITT CO data, their relative rates of decline were very different. Compared with the base year of 2006, the surface CO declined by 1.6% annually and MOPITT CO declined by 0.4% (in mixing ratio) and 0.6% (in total column), respectively. Much more decline percent in surface measurement than satellite result indicated different influencing factors on surface and total column CO. **"We found significant correlations ($p < 0.05$) between surface CO and MOPITT CO ($r = 0.75$ and $0.61$ for the MOPITT CO mixing ratio and total column, respectively) data (see Fig. S4), which indicate the good regional representativeness of Lin'an measurements."** was added in 3.5.4 Implications on regional atmospheric chemistry in the revised manuscript.

[Figure]

Fig. S1. Correlations between annual surface CO measured at the LAN station and the MOPITT CO data from 2006 to 2017

Diurnal cycle: the PBL mixing would be an important factor affecting the diurnal cycle. Could you show PBLH or other indicator of mixing (e.g., from assimilation data)? Also, there appears to be some phase shift in the peak mixing ratio between different sites. Could you comment on this?

Response: Yes, the PBL mixing would be an important factor affecting the diurnal cycle. Here, we used Hysplit model from NOAA and NCEP meteorological data to calculate the Planetary Boundary Layer Height (PBLH) at the LAN station and its three surrounding cities in January, April, July, and October. The Average diurnal variations in the Planetary Boundary Layer Height (PBLH) at Shanghai, Nanjing, Hangzhou, and the LAN station in 2019 were plotted in Fig. S2. Fig. S3 also presented the PBLHs calculated by a high-resolution Planetary Boundary Layer Parameterization (MM5 Blackadar Scheme) in 2019 at 4 sites. The PBLH was rather high during the daytime and usually reached its highest around 14:00, which indicated that the pollutants in the atmosphere were well mixed in the afternoon and corresponded to the time when the lowest CO mixing ratios were observed (Fig. 4. in the manuscript). Since the diurnal variations in the PBLHs at 4 sites were almost the similar according to the hourly resolution, the little phase shift in the CO mixing ratio peak between different sites was likely attributed to the difference in local emissions. "**The Planetary Boundary Layer Height (PBLH) is a key indicator of atmospheric mixing state. As shown in Fig. S3 and S4, the PBLH was rather high during the daytime and usually reached its highest around 14:00, which indicated that the pollutants in the atmosphere were well mixed in the afternoon and corresponded to the time when the lowest CO mixing ratios were observed (Fig. 4). Since the diurnal variations in the PBLHs at 4 sites were almost the similar according to the hourly resolution, the little phase shift in the CO mixing ratio peak between different sites was likely attributed to the difference in local emissions.**" were added to the section 3.3 Diurnal variation in the manuscript.

[Figure]

Fig. S2. Average diurnal variationsin the Planetary Boundary Layer Height (PBLH) at Shanghai, Nanjing, Hangzhou, and the LAN station in (a) January, (b) April, (c) July, and (d) October, 2019. PBLH calculated by Hysplit model and NCEP meteorological data.

[Figure]

Fig. S3. Average diurnal variations in PBLH at Shanghai, Nanjing, Hangzhou, and the LAN station in 2019. PBLHs calculated by a high resolution Planetary Boundary Layer Parameterization (MM5 Blackadar Scheme), which were provided by Doctor Liu Hongli from the Chinese Academy of Meteorological Sciecnes.

Interannual variability: The variability is very large for CO mixing ratios, which may be related to

meteorology and/or chemistry. Could you comment on this? What does this variability mean when relating CO concentrations to emissions?

Response: This is a good but complex question. In section 3.5.2, we try to demonstrate that the meteorology was not the main factor contributing to the descending trend of CO. As for the interannual variability, it should be influenced by many factors and sometime it's hard to specify which one was the most important. Here, a simple method was used to answer this question. In this method, the differences of meteorological elements (T, P, RH, WS), CO, $O_3$, and average emissions in adjacent two years were calculated, and then their correlations were made and showed in Table S1. Here, $O_3$ was used as the indictor of the oxidizing capacity (chemistry).

As indicated in Table S1, the interannual variability of CO showed no significant correlation with the fluctuations of any other single element, which suggested that the annual variability of CO was caused by a comprehensive impact from all the factors.

Table S1. Correlation between different meteorological elements (T, P, RH, WS), CO, $O_3$, and average emissions in adjacent two years (p-value were shown in the brackets)

|  | ΔT | ΔP | ΔWS | ΔRH | ΔCO | ΔO3 | ΔEmissions |
|---|---|---|---|---|---|---|---|
| ΔT | 1.00 | -0.57 (0.07) | 0.50 (0.12) | -0.50 (0.12) | -0.53 (0.10) | -0.02 (0.96) | 0.28 (0.41) |
| ΔP |  | 1.00 | 0.03 (0.93) | -0.09 (0.78) | 0.08 (0.82) | -0.13 (0.71) | -0.08 (0.82) |
| ΔWS |  |  | 1.00 | -0.18 (0.60) | -0.27 (0.42) | -0.13 (0.69) | 0.30 (0.37) |
| ΔRH |  |  |  | 1.00 | 0.11 (0.75) | 0.31 (0.36) | -0.34 (0.30) |
| ΔCO |  |  |  |  | 1.00 | 0.02 (0.95) | 0.05 (0.88) |
| ΔO3 |  |  |  |  |  | 1.00 | 0.02 (0.95) |
| ΔEmissions |  |  |  |  |  |  | 1.00 |

To what extent is the trend of CO consistent with those of emissions, quantitatively? And why?

Response: We conducted a correlation analysis between the annual mean CO concentrations and the anthropogenic emissions of CO in the neighboring provinces, and we found they exhibited a strong positive correlation (r = 0.83, p<0.01). Also, compared with the base year of 2006, the CO concentration in 2017 declined by 18.7 %, which is close to the decline value of 31.3% for the average anthropogenic emissions of CO in the neighboring provinces. The decreasing percentage of the CO concentrations and the emissions were overall consistent when considering larger uncertainty existing in emission data. Therefore, the declined trend of CO at the LAN station was mainly attributed to the cut-down of anthropogenic emissions in the YRD region. "**There was a strong positive correlation (r = 0.83, p<0.01) between the annual mean CO concentrations and the anthropogenic emissions of CO in the neighboring provinces. Also, compared with the base year of 2006, the CO concentration in 2017 declined by** 18.7**%, which is close to the decline value of** 31.3**% for the average anthropogenic emissions of CO in the neighboring provinces. The decreasing percentage of the CO concentrations and the emissions were overall consistent when considering larger uncertainty existing in emission. Therefore, the declined trend of CO at the LAN station might be mainly attributed to the cut-down of anthropogenic emissions in the YRD region**." were added to the section 3.5.3 Changes in emissions in neighboring provinces in the manuscript.

Please define alpha for the first time (I assume it is the P-value). Also, why is alpha always equal to

0.05 throughout the text? Do you mean alpha < 0.05? Specifying the actual value of alpha would be better, given that the often mis-interpretation of the P-value, as discussed extensively in recent years.
Response: In this paper, alpha=0.05, means the statistics was significant at the confidence level of 0.05. We use the P values instead of α in the revised paper.

Fig. 6. The choice of the three periods seems to be arbitrary – it is not obvious why the years can be grouped into these three periods. Could you show Lorenz curves year by year?
Response: Thanks, as was mentioned in the manuscript, considering the annual variation trend and the major air pollution control policies adopted during the study period, we divided the study data into three subsets of data. Grouped data could reflect clearer difference of the distribution than year by year. The Lorenz curves year by year which were plotted in Fig. S4. The CO concentration with the highest frequency in 2010 was significantly lower than that from 2006 to 2009, and the peak CO mixing ratios experienced a sharp drop in 2016 and 2017 (below 0.50 ppm) compared to those in the previous years. Therefore, we grouped the data into 3 sub datasets.

[Figure]

Fig. S4. Frequency distribution of the CO mixing ratios and Lorentz curve fitting from 2006 to 2017.

Fig. 8. There appears to be some non-zero trend in the 5% percentile data. Could you show the trend and discuss this in the context of the large-scale CO trend (e.g., shown in the literature)?
Response: We added the p-value and r in Fig. 8. The CO concentrations at the 5th percentiles exhibited a negative correlation with the change of years, but it was not statistically significant (r = -0.41, p = 0.19). The decreasing trend at 5% was to some extent in accordance with the large scale declined trend of CO around the world (see informations in **Introduction).** Sentences in the manuscript were revised as "**the CO concentration under the polluted condition experienced a significant decreasing trend of -22.4 ppb/yr (r = −0.68, α = 0.05) from 2006 (0.91 ppm) to 2017 (0.58 ppm) and that under the clean condition descended as well but not statistically significant**

**(r = −0.41, p = 0.19) throughout the years. This suggests that the CO levels in pollution plumes, which are highly impacted by anthropogenic emissions in the YRD region, have been reduced greatly, and the background levels of CO at the LAN station also showed a decreased evidence at the same time.**"

Fig. 9. The value in 2011 is also very low. Could you explain this?

Response: As suggested in the previous question, the interannual variability sometime was very tricky. Since the meteorological conditions like temperature, air pressure, wind speed, relative humidity, and the wind direction frequency in 2011 were not changed obviously, the low value in 2011 should be due to the the continued influence of the air pollution control measures taken during the Shanghai World Expo in 2010. This can be supported by the conclusion of better air quality in 2010 and 2011 in Shanghai than the previous years in Zhen et al. (2012).

Fig. 10. The analysis for Fig. 10 appears speculative. The apparent relationships are a result of processes at a variety of time scales, which render determinative explanations difficult. Also, the met fields used are correlated to each other, affecting the explanations specific to each met field.

Response: When comparing the CO levels in different time periods to evaluate the effectiveness on emission reduction, the influence by the different meteorology conditions is a topic that can't be bypassed. To minimize the meteorological effects, the data in 3 periods were reclassified according to temperature, RH, wind speed and wind direction (Fig. 10). Although the meteorological parameters are correlated to each other, this kind of control variate method could provid a similar base for comparisons. The results further proved the declined trend was due to CO emission reduction.

Specific comments: L169, should be 12 years L526. "those" should read "than those" Table 1. The mixing ratio at Shangdianzi appears very low in 2011-2017. Could you check this? Fig. 3b. Do you mean the number of fire spots here? Fig. 5. Is the standard deviation calculated from monthly means? Fig. 7. Is weighting factors based on standard deviation of monthly mean values? Please clarify. Fig. 10 and Table 2. Please specify the intervals of each met field.

Response: Thanks, all of these words were revised.

The mixing ratio at the Shangdianzi was indeed very low reported by (Liu et al., 2019), because the CO values were determined after removing those impacted by local anthropogenic emissions using the technique of robust extraction of baseline signal (also noted in the manuscript).

Yes, we mean the number of fire spots and the y-axis label has been revised.

Yes, we have added the calculation of standard deviation in Fig.5 caption in the revised manuscript as "**the error bars represent one standard deviation calculated from monthly means.**"

Yes, we have noted in the revised manuscript as "**with standard deviation of monthly mean values being used as weighting factors**."

The intervals of each met field used in Fig.10 were 5℃, 0.5 m/s, 10%, and 22.5º for T, WS, RH, and WD, respectively. The Fig. 10 caption was also revised as "**Fig. 10. Variations of CO mixing ratios in different periods with respect to temperature (T), Wind Speed (WS), Relative Humidity (RH), and Wind Direction (WD). The intervals are 5℃, 0.5 m/s, 10%, and 22.5º for T, WS, RH, and WD, respectively**."

The decreased percentage shown in Table 2 was calculated from the average CO concentrations

under each met field in Fig. 10.

**References**

Liu, S., Fang, S. X., Liang, M., Sun, W. Q., and Feng, Z. Z.: Temporal patterns and source regions of atmospheric carbon monoxide at two background stations in China, Atmos Res, 220, 169-180, https://doi.org/10.1016/j.atmosres.2019.01.017, 2019.

Zhen, X. R., Chen, L., Mao, Z. C., and Ma, L. M.: Review on Air Qualities over Shanghai in 2011. Atmospheric Science Research and Application, 1, 51-60, 2012.